# Ras Pathway Activation and MEKi Resistance Scores Predict the Efficiency of MEKi and SRCi Combination to Induce Apoptosis in Colorectal Cancer

**DOI:** 10.3390/cancers14061451

**Published:** 2022-03-11

**Authors:** Thomas Benjamin Davis, Shilpa Gupta, Mingli Yang, Lance Pflieger, Malini Rajan, Heiman Wang, Ramya Thota, Timothy J. Yeatman, Warren Jackson Pledger

**Affiliations:** 1Department of Surgery, University of Utah, Salt Lake City, UT 84132, USA; u0878193@utah.edu (S.G.); mingli.yang@utah.edu (M.Y.); mrajan@u2m2.utah.edu (M.R.); heiman.wang@utah.edu (H.W.); yeatman7383@gmail.com (T.J.Y.); 2Precision Genomics Translational Science Center, Intermountain Healthcare, Murray, UT 84107, USA; lance.pflieger@imail.org; 3Oncology Clinical Program, Intermountain Healthcare, Murray, UT 84107, USA; ramya.thota@imail.org; 4Huntsman Cancer Institute, University of Utah, Salt Lake City, UT 84112, USA

**Keywords:** colorectal cancer (CRC), drug resistance, biomarkers, drugs sensitivity selection, MEK inhibition, SRC inhibition, gene expression, gene signature score, consensus molecular subtype (CMS)

## Abstract

**Simple Summary:**

Inhibition of MEK has been proposed as a means to address mutant RAS colorectal cancer (CRC). However, MEK inhibitor adaptive resistance has led to reduced clinical utility of this new drug. Our studies have suggested the potential for the addition of an SRC inhibitor to prevent the development of resistance to MEK inhibitors. Moreover, we have identified that gene expression signature scores for RAS pathway activation, and MEK inhibitor resistance may be useful biomarkers in determining CRC drug sensitivity to the novel combination of Trametinib and Dasatinib.

**Abstract:**

Colorectal cancer (CRC) is the second leading cause of cancer death in the United States. The RAS pathway is activated in more than 55% of CRC and has been targeted for therapeutic intervention with MEK inhibitors. Unfortunately, many patients have de novo resistance, or can develop resistance to this new class of drugs. We have hypothesized that much of this resistance may pass through SRC as a common signal transduction node, and that inhibition of SRC may suppress MEK inhibition resistance mechanisms. CRC tumors of the Consensus Molecular Subtype (CMS) 4, enriched in stem cells, are difficult to successfully treat and have been suggested to evade traditional chemotherapy agents through resistance mechanisms. Here, we evaluate targeting two pathways simultaneously to produce an effective treatment by overcoming resistance. We show that combining Trametinib (MEKi) with Dasatinib (SRCi) provides enhanced cell death in 8 of the 16 tested CRC cell lines compared to treatment with either agent alone. To be able to select sensitive cells, we simultaneously evaluated a validated 18-gene RAS pathway activation signature score along with a 13-gene MEKi resistance signature score, which we hypothesize predict tumor sensitivity to this dual targeted therapy. We found the cell lines that were sensitive to the dual treatment were predominantly CMS4 and had both a high 18-gene and a high 13-gene score, suggesting these cell lines had potential for de novo MEKi sensitivity but were subject to the rapid development of MEKi resistance. The 13-gene score is highly correlated to a score for SRC activation, suggesting resistance is dependent on SRC. Our data show that gene expression signature scores for RAS pathway activation and for MEKi resistance may be useful in determining which CRC tumors will respond to the novel drug combination of MEKi and SRCi.

## 1. Introduction

Although colorectal cancer (CRC) is one of the leading causes of cancer deaths in the United States, it can be surgically cured with early detection. Nevertheless, many CRC tumors are not diagnosed until later stages when cure rates are low (15% five-year survival) [1]. The Wnt pathway is altered in over 80% of CRC most frequently because of mutations in APC [2]. Mutationally activated KRAS, BRAF or NRAS, which drive the oncogenic RAS/RAF/MEK/ERK pathway, are present in 55% of CRC tumors and portend poor survival [3,4,5,6]. The PI3K/AKT pathway is activated in more than 20–40% of CRC [7]. Even though targeted therapies have been developed with excellent specificities for specific components of driver pathways, clinical responses have been disappointing. Thus, effective inhibitors of EGFR, BRAF, MEK, AKT and ERK have not proven clinically efficacious, most likely due to drug resistance, and the inability to identify de novo drug sensitive tumors [8,9,10,11,12]. Clearly, adaptive drug resistance and the presence of more than one oncogenic pathway in CRC tumors could limit the success of targeted therapies [8,9,12,13].

SRC signaling may be largely responsible for drug resistance to targeted inhibition of the RAS/MEK/ERK. It is known that SRC activity and expression is increased in CRC tumors as compared to the SRC activity found in normal colon tissue [14,15,16] and that SRC activity increases with CRC tumor progression and metastasis [17]. A high expression level of SRC is associated with decreased survival in patients with CRC [18]. Furthermore, in ovarian cancers the limited response to MEK inhibitors (MEKi) has been suggested to be related to resistance due to SRC activity [19]. Park et al., using a Boolean network simulation, confirmed our previous results that SRC inhibition could reduce adaptive resistance, allowing greater sensitivity to MEKi in CRC [20,21]. In vivo studies for multiple solid tumors have demonstrated that SRC activity may also be responsible for resistance to chemotherapy [22,23]. We hypothesized that SRC may represent a common node that is responsible for adaptive resistance to MEK/ERK pathway inhibition. Furthermore, the consensus molecular subtypes (CMS1-4) were recently created from a comprehensive, unsupervised molecular analysis of thousands (*n* = 4151) of human tumors to best define CRC [24,25]. SRC activation has been linked to the CMS4 class. This subtype accounts for 23% of all CRC, with 40% being KRAS mutants [24]. CMS4 CRCs are notable for being difficult to treat with conventional chemotherapies [26]. Even though other cellular mechanisms upregulating different bypass pathways may contribute to drug resistance, our investigations led us to focus on two pathways to overcome drug resistance and allow efficacious therapies. These two pathways are the EGF/RAS/MEK/ERK pathway and SRC family pathway.

Here, we studied an 18-gene RAS pathway activation score and a 13-gene MEKi resistance score. Notably the 13-gene score predicts drug resistance caused by “bypass” proliferation/survival signaling pathways in the presence of active MEK but does not depend on PI3K activity [27]. Therefore, we have not included the PI3K/AKT/mTOR pathway in our studies. The 13-gene MEKi resistance score, however, does correlate with both a Dasatinib sensitivity score and an SRC activation score [28,29]. Due to the striking correlation of the 13-gene MEKi resistance score and SRC activation score, we thought that SRCi may be a means to overcome MEKi resistance. We demonstrate that pathway signature scores for the 18-gene RAS pathway signature and the 13-gene bypass MEKi resistance signature score serve as biomarkers for CRC cells, especially the CMS4 class, that are sensitive to the therapeutic combination of MEKi and SRCi. These data provide a predicate for future in vivo and clinical studies re-purposing this drug combination.

## 2. Materials and Methods

### 2.1. Cell Culture

HCT116, SW48, WiDr, SW1417, SNU-C2B, SNU-2CA, and SW480 were sourced from the American Type Culture Collection (ATCC). LIM2099 was purchased from MilliporeSigma (Burlington, MA, USA). SW837 was sourced from Amgen (Thousand Oaks, CA, USA), DiFi was donated from Robert Coffey (Vanderbilt University Medical Center Nashville, TN, USA), RW7213 was donated from Sanjay Goel’s lab (Montefiore Medical Park, NY, USA) and SNU-1411 was donated from Sandra Misale’s lab (Sloan Kettering Institute, NY, USA). All cell lines had biweekly tests for mycoplasma contamination with MycoAlert Mycoplasma Detection Kit from Lonza Walkersville, Inc. (LT07-418 Walkersville, MD, USA). Cell lines were cultured using RPMI 1640 supplemented with 10% FBS and 1% penicillin and streptomycin from Gibco (Waltham, MA, USA).

### 2.2. Immunoblotting

Cells were lysed using 1×RIPA buffer (9806 Cell Signaling Danvers, MA, USA) containing 10 µM PMSF, Phosphatase Inhibitor Cocktail 2 (P5726 MilliporeSigma), Protease Inhibitor Cocktail (M250 Amresco Dallas, TX, USA), and Phosphatase Inhibitor Cocktail 3 (P0044 MilliporeSigma). Protein samples were run on 4–15% Mini-PROTEAN TGX Precast Protein Gels from Bio-Rad (4561086 Hercules, CA, USA). The LI-COR Odyssey^®^ CLx Imaging System was used for imaging all immunoblots (LI-COR Lincoln, NE, USA). Antibodies were typically duplexed using Li-Cor secondary antibodies, Goat anti-Rabbit IRDye 680RD and Goat anti-Mouse IRDye 800CW. Rabbit primary antibodies were used unless specified and were obtained from Cell Signaling: phospho-Erk1/2 T202/Y204 (Cat. No. 4370); Cleaved PARP (Asp214) (D64E10); phosphor-SRC Y416 (2101) and SRC (mouse 2105).

Band density was determined using LI-COR’s Image Studio™. Density readings for Phospho-ERK and Phospho-SRC were normalized using the density readings of the total protein for each respectively. Cleaved PARP density readings were normalized using readings for ß-Actin. All westerns were performed in triplicate. Normalized band density was then averaged, and standard deviation was calculated. The *p*-values for normalized band density were determined using the Student’s *t*-test.

### 2.3. Cell Viability Assays

Cells were plated in white opaque Corning™ 96-Well, Cell Culture-Treated, Flat-Bottom Microplate (cat. 07200628). Following 24 h of growth, cells were then treated with the inhibitors at various dilutions ranging from 0.01–1 µM and all the permutation combinations of Trametinib and Dasatinib. After appropriate hours of treatment, the Promega Cell Titer-Glo^®^ 2.0 Cell Viability Assay (G9241) was performed to determine cell viability. Plates were read using the Thermo Fisher Scientific (Waltham, MA, USA) Varioskan LUX Multimode Microplate Reader.

### 2.4. Inhibitors and Apoptosis Assay

All CRC cell lines were treated with Trametinib (HY-10999) or Dasatinib (HY-10181) sourced from Medchem Express. Treatments ranged from 48–96 h as noted in each assay. For all assays, cells were plated and allowed to grow for 24 h, then treated with inhibitors. Following treatments, Annexin V apoptosis assays were performed as described previously [21,30]. The analyses were performed with a BD FACSAria™ Fusion Flow Cytometer.

### 2.5. Gene Signature Score Analysis and CMS Classification

The 18-gene MEK pathway activation, 13-gene MEKi resistance, and SRC activation signatures were adopted from previous analyses reported from other groups [27,28]. Of note, Broecker et al. reported a transcriptional signature induced by the metastasis promoting SRC-mutant that activated SRC signaling in breast cancer [28]. Here we adopted their 61 up-regulated genes to calculate the SRC activations score for cell lines and tumors. Gene signature scores were generated using the Gene Set Variation Analysis R package with the ssgsea method [31].

For CMS classification, CRC cell line or tumors were classified by CMScaller (an R package for consensus molecular subtyping of colorectal cancer pre-clinical models) as described by Eide et al. [32].

For the Medico et al. CRC cell line dataset [33], we used their global gene expression data for 18-gene and 13-gene signature score analysis and for CMS classification. Affymetrix gene expression data of 155 cell lines were downloaded via GEO with accession number GSE59857. Note that CO115 (that was established from a tumor implanted into nude mice) was excluded from analysis. Notably, 118 cell lines (out of 155) were classified as CMS1-4 cell lines for further analysis.

For human colorectal tumor validation, we used the Clinical Proteomic Tumor Analysis Consortium (CPTAC) dataset [34]. RNA expression values from the CPTAC study were downloaded from the cBioportal repository (https://www.cbioportal.org/datasets, accessed on 5 October 2021). The downloaded expression values were normalized by the version 2 cBioportal pipeline (RSEM upper quartile Log2(*n* + 1)). A total of 101 of the samples were significant classified into a CMS subtype and used in this analysis.

The signature scores were calculated as previously described [35,36]. For the Medico 154 CRC cell lines, we note that probe values of some signature genes appeared to differ from one another by a few orders of magnitude. To avoid over-representation of only a few dominant probes or genes in calculating signature scores, individual probe values of a signature gene were first normalized by the mean of all 154 cell lines. Next, the mean of the probe values was calculated for each signature gene. A signature score for each cell line was calculated by averaging all gene values. Finally, scores were standardized by subtracting the score median and dividing by the score IQR (interquartile range) of all 154 cell lines.

### 2.6. Statistical Analysis

Cell culture experiments were done in triplicates, and mean, median and standard deviation were calculated as indicated. A one-tailed, paired t test was used to determine the statistical significance of comparison as needed. Correlation analysis, t test and Chi square trend test as well as CMS classification were performed using GraphPad Prism version 8.00 (La Jolla, CA, USA) and R version 3.6.2.

## 3. Results

### 3.1. CRC Cell Line Selection and Associated MEK and SRC Activation

The CRC cell lines listed in Table 1 were used to determine if MEKi alone is effective in promoting cell death in CRC cell lines, and if the combination with SRCi has the capacity to bring about greater cell death than either inhibitor alone. These cell lines were selected to represent CMS classes. Thus, 9 CMS4 cell lines were selected with an additional 2 CMS1, 2 CSM2, and 3 CSM3 class cell lines for a total of 16 cell lines. The data in Figure 1 compare the level of MEK and SRC activation in each of these CRC cell lines. The amount of MEK and SRC activation was determined using phospho-tyrosine/serine Western Blot analysis (full blots featured in Appendix A). This comparison was performed to determine if the canonical increase in SRC activity for CMS4 cell lines is reflected in their levels of phosphorylated SRC tyrosine 416 [30]. As we can see, the SRC activation does not seem inherently contingent on the CMS class.

### 3.2. The Effect of Inhibition of MEK and SRC on Cell Viability

We determined the change in cell viability caused by the inhibition of MEK activity using Trametinib (MEKi) in cultured CRC cell lines (Table 1). Cells were grown and treated with Trametinib at the indicated concentrations for 96 h (Figure 2). Using the CellTiter-Glo Luminescent cell viability assay-Promega (CellTiter-Glo) we found that 0.1µM of Trametinib had maximum effectiveness on cell viability and this inhibition was similar to the effect of ERK inhibition [21]. Thus, in agreement with other published studies, Trametinib at 0.1 µM proved to be an effective concentration to inhibit cell viability in these cell lines [21]. We next determined the effect on cell viability caused by the inhibition of SRC activity with Dasatinib (SRCi). Cells were cultured in the presence of various concentrations of Dasatinib, ranging from 0.01 µM to 1.0 µM, for 96 h (Figure 2). Our data show that 0.1 µM of Dasatinib is an effective concentration for reducing cell viability (CellTiter-Glo) during the growth of the CRC cell lines (Table 1). We selected 0.1 µM as an appropriate dosage of Dasatinib to use in order to determine if the combination of MEK inhibition with SRC inhibition had greater effectiveness than either drug alone. Additionally, the Dasatinib concentration of 0.1 µM is relevant to the clinical dosage [37]. Various concentrations of Trametinib with Dasatinib were used in combination to treat the CRC cell lines. The cells were treated with each drug alone or in combination for 96 h and then the cell viability was determined with the CellTiter-Glo assay. Data for each condition for these cell lines are illustrated in Appendix A, but the cell viability for each cell line subjected to 0.1 µM Dasatinib and 0.1 µM Trametinib and the two drugs in combination are shown in Figure 3. These data point out that the combination of MEK and SRC inhibition displayed a greater effect on cell viability in some cell lines compared to a single agent. However, our data show Trametinib to be the major contributing factor to the inhibition in most of these cell lines. We see a significant decrease in viability with two of the cell lines, LIM2099 and LS123. However, we point out that the CellTiter-Glo assay represents changes in cell viability that may depend on stimulated cell death and the cytostatic inhibition of cell growth.

### 3.3. Drug Stimulated Apoptosis

We suggest that cell viability might not be the best measure for determining drug sensitivity when the effect of a strong growth suppressor, such as a MEK inhibitor, is measured. Thus, we sought to determine the effectiveness of the combination of the MEK inhibitor with SRC inhibition directly on apoptosis. The cell lines (Table 1) were grown in standard culture conditions and treated with 0.1 µM Trametinib or 0.1 µM Dasatinib as single agents or with the two drugs in combination for 96 h of treatment. All treatments were performed in triplicate and assayed for cell death, using an ANNEXIN V Assay. Figure 4 shows the apoptosis induced in each cell line when treated with Trametinib or Dasatinib alone and the combination of both drugs. We also investigated the inhibition of MEK and SRC activities with 0.1 µM Trametinib and 0.1 µM Dasatinib shown in Figure 5. The induction of PARP cleavage in each cell line treated is also shown in Figure 5. When either MEK activity or SRC activity were inhibited, most cell lines showed some induction of apoptosis (Figure 4) and increased PARP cleavage (Figure 5); however, for many of the CMS4 cell lines the maximum PARP cleavage and induction of Annexin occurred when both MEK and SRC activities were inhibited (full blots available in Appendix A).

### 3.4. Signature Scores to Predict Drug Sensitivity

We hypothesized a subgroup of CRC tumors may be sensitive to treatment with the combination of MEK and SRC inhibitors and can be selected utilizing gene expression signatures translated into quantitative biomarker scores. We have shown that scores measuring pathway activation and drug sensitivity can be useful to identify subpopulations of patients whose tumors are sensitive to a drug therapy [21]. Recently, an 18-gene RAS pathway signature was developed representing the 18 genes whose expression changed due to KRAS expression. The score is determined by the total change in expression of the 18 genes, either increased or decreased numerical values [27]. The higher the RAS score, the greater was the predicted sensitivity to pathway inhibitors such as MEK inhibition. However, many cell lines are resistant to MEK inhibition and a 13-gene signature was determined that correlated to MEKi resistance. The summation of the amount of the change in each of the 13 genes’ expression, whether positive or negative, together yielded the MEKi resistance bypass score [27].

CMS4 tumors are most resistant to chemotherapy, and our hypothesis predicts they are sensitive to the combination of MEK and SRC inhibition. The CMS 1 group is rich in MSI tumors. We hypothesized that cell lines with a high RAS pathway score and a high 13 gene MEKi resistant score wound be sensitive to the combination of Trametinib and Dasatinib. In order to test our hypothesis, we used the gene expression scores of the cell lines in Table 1.

Figure 6 displays the CRC cell lines used in this study plotted by their 18-gene signature scores and their 13-gene MEKi resistant scores. We had predicted the cell lines in the right upper quadrant would have high sensitivity to MEK inhibition, but with high MEKi resistance scores correlating with high SRC activation, the combination of MEKi and SRCi might bring about enhanced apoptosis. When we compared the drug sensitivities of the tested cell lines (Figure 4) with their 18-gene and 13-gene scores seen in Figure 6, we found that the two signature scores predict cell lines that had enhanced or additive MEKi plus SRCi-induced cell death. The eight cell lines marked with checkerboard pattern with higher scores in the upper right quadrant display enhanced or additive drug sensitivity when used together. This increase in sensitivity is shown to be significant in the Annexin V data from Figure 4. Notably, all the cell lines in the upper right-hand quadrant with enhanced sensitivity to the combination drug treatment were identified as CMS4 class (Table 1), and only one, SW837, of the CMS4 cell lines was not sensitive to the drug combination.

To further support the hypothesis suggested by our data, we used the Medico cell line database and plotted 118 CMS1-4 CRC cell lines with their 18-gene RAS signature scores and their 13-gene MEKi resistance scores (Figure 7a). Interestingly, the distribution patterns of CMS1-4 cell lines with lower 13 gene scores and ones with higher 13 gene scores appear distinct. A Chi-square test with a value of 72.9 (*p* < 0.0001) for these cell lines shows a distinctive trend for CMS4 cell lines to have higher 18-gene scores and higher 13-gene scores, which fall in the right upper quadrant (RUQ) (Figure 7a). Notably, when cell line distributions are analyzed in each CMS class based by quadrants (Figure 7b), nearly all CMS4 cell lines have higher 13-gene scores (locating within primarily RUQ and secondary LUQ). To a lesser degree, the CMS1 cell lines also tended to have higher 13-gene scores. By contrast, the CMS2 and CMS3 cell lines tended to have lower 13-gene scores, falling in the lower two quadrants (LLQ and RLQ).

Next, we extended our in vitro and in silico cell line experiments to human tumors. Figure 8 displays the relationship of the 18-gene RAS signature score and the 13-gene MEK resistance score for 101 CRC tumors in the Vasaikar database [34]. We found a unique separation when tumors were plotted using the 18-gene RAS pathway activation score and the 13-gene MEKi resistance score in a large CRC tumor database. We hypothesized that tumors in the upper right quadrant are sensitive to a combination of MEKi with SRCi. Dividing the tumors into CMS subpopulations, these tumors plotted by the 18-gene Ras signature score and the 13-gene MEKi resistance score revealed the CMS4 tumors to have higher 18-gene and 13-gene scores (Figure 9), which matches the same pattern seen in the cell line data (Figure 7). In order to explain this observation of drug sensitivity, we analyzed the CRC human tumor database similarly as earlier in Figure 6 to compare tumors by plotting the 13-gene MEKi resistance score and an SRC activation score (Figure 10). This figure clearly points out the close correlation of 13-gene MEKi resistance and SRC activation in CRC tumors and suggests the MEKi resistance could be largely dependent on SRC activity.

## 4. Discussion

We have demonstrated that the combination of a MEKi and SRCi can effectively kill many CRC cell lines (Figure 5), whereas few are sensitive to either drug alone. Neither drug by itself has proven significantly effective in treating human CRC; however, the combination of the two drugs has never been attempted clinically. Thus, we set out to develop the hypothesis that the combination of MEKi and SRCi might be effective in certain CRC subtypes that could be identified with novel biomarkers. Cell lines that are sensitive to the combination of MEKi and SRCi cannot be selected based on the apparent amounts of activated MEK and SRC. For example, the activity of the cells shown in Figure 1 does not allow the prediction of apoptosis seen in Figure 4 due to MEK and SRC inhibition. The SRC activity in WiDr is lower than that of LIM2099 or LS123, yet Dasatinib brings about as much apoptosis in WiDr as in LIM2099 or LS123. This same enhancement is seen with the combination of MEKi and SRCi in WiDr and SW480 when compared to LIM2099 and LS123. For these cells, the data in Figure 4 illustrate that the MEKi plus SRCi had enhanced cell death as compared to either inhibitor used as a single agent. Interestingly, the comparison of the results in Figure 3 reporting CellTiter-Glo data is not equal to the cell death as that found when Annexin V staining was used to measure cell death (Figure 4). This comparison is not surprising since the CellTiter-Glo assay measures cell viability. Therefore, when the control untreated culture is grown and compared to growth with a drug that blocks cell growth, the assays using cell viability may be reflecting growth inhibition along with cell death. Our studies measuring cell death were performed with Annexin V staining (Figure 4) and PARP cleavage (Figure 5). The data on enhanced/additive apoptosis with the two-drug combination of MEKi and SRCi agreed with our data for HCT116 cells using ERKi with an SRCi. Similar conclusions regarding the use of SRCi with a MEKi were reached in ovarian and CRC cell lines confirming our current and previous observations [19,20,21]. Since our data indicated that half of the tested CRC cell lines responded to the combination of MEKi and SRCi, it would be significant to predict which cell lines of CRC tumors are sensitive to this drug combination. Thus, therapeutic response rates determined for the sensitive CRC population of CRC tumors would be much greater than the total CRC tumor response.

We have pioneered the use of large CRC databases with DNA sequencing and global RNA expression to determine tumors that are sensitive to specific drugs and to discover genes that are associated with tumors and tumor progression. For example, we used a validated RNA gene expression signature that reflects cetuximab sensitivity [4,21] and determined cetuximab sensitive scores are greater in tumors that have mutations in APC and p53 [21]. Although this seems contrary to accepted treatment protocols for CRC tumors with KRAS, it is now under investigation with a U.S. National Cancer Institute (NCI) supported clinical trial.

Using a validated 18-gene RAS signature score that measures response to KRAS pathway activation, we explored a CRC database to search for genes that altered the KRAS pathway signature. We documented that mutations in PTPRS brought about increased RAS pathway activation [38]. Our published data demonstrated that PTPRS could affect the RAS pathway by its direct action on ERK phosphorylation. Interestingly, we found that PTPRS mutations occur in 10% of CRC tumors in our database and also in the Dana Farber CRC tumor database [5].

Here we used signature scores to explore drug response to determine predictive biomarkers for drug sensitivity. The signature score represents the numerical value of all the genes altered in the signature. For example, the score of the 18-gene RAS pathway signature is the expression values of all eighteen upregulated genes. This KRAS signature score is indicative of the tumor’s expression of the RAS/MEK/ERK pathway. Dry et al. [27] developed a 13-gene MEK resistance signature that did not include PI3K or AKT activities. Such tumors could be sensitive to MEKi, but most are not. However, a high 13-gene MEKi resistance score was determined to influence the sensitivity to RAS pathway targeted therapies. Tumors plotted with the 18-gene KRAS pathway activation score and the 13-gene MEKi resistance score suggest many of the tumors with highest RAS signature scores had the highest 13-gene resistance signature score (Figure 6). We demonstrated the 13-gene resistance score had a close correlation with a validated SRC activation score, suggesting SRC activity may be responsible for the MEKi resistance (Figure 10). We hypothesize that tumors with a high 18-gene signature score and a high 13-gene MEKi resistance score will be sensitive to the combination of a MEKi with an SRCi. The comparison of the CRC cell lines we used (Table 1) with the 18-gene RAS pathway score and the MEKi resistance score displayed 8 of the cell lines in the upper right quadrant in Figure 10. We noted that these cell lines were sensitive to the combined treatment of MEKi and SRCi. The cell lines with lower scores for either the 18-gene or 13-gene signature scores show resistance to MEK and SRC inhibition in a predictive manner. RW7213 and SNU1411 both have positive 18-gene scores while having a low 13-gene score. This is reflected in both having sensitivity to Trametinib but not Dasatinib and the combination having no significant effect on increasing apoptosis. Thus, these cell lines probably have a mechanism of resistance separate from or in addition to SRC activity. Furthermore SNU-2CB, DiFi, and SW48 all have low 18-gene scores and show heavy resistance towards apoptosis when Trametinib alone is used. However, cell lines SW837, SW620 and LoVo do not fit their gene scores as precisely as the others. The combination sensitive cell lines all have in common positive 18-gene and 13-gene scores.

The apoptosis data and cell viability data do suggest that a single gene score for 18-gene or 13-gene is not inherently an accurate predictor of sensitivity for these cell lines in regard to cell death. Cell lines such as LIM2099, LS123, SNU-2CB and SNU-2CA fit well with their higher 13-gene scores, representing MEKi resistance. However other cell lines with higher 13-gene scores such as SW480, HCA-7, HCT116, and WiDr all show some amount of sensitivity to Trametinib alone in terms of apoptosis. However, with the 13-gene score we did find as a strong predictive score for MEKi and SCRi combination sensitivity.

We believe this to be highly clinically relevant as the MEKi and SRCi combination sensitive cell lines were all shown to be CMS4 cells. It should also be noted that our plotting of tumor and cell line expression data revealed that CMS4 class tumors and cell lines are more inclined to have higher 18-gene and 13-gene scores than the other CMS classes. This finding is clinically relevant as CMS4 tumors have been shown to be chemotherapy-resistant and have a worse prognosis. Moreover, this class is enriched with cancer associated fibroblasts and cancer stem cells. Our data are consistent with a recent report suggesting that MEKi activate WNT signaling by downregulating AXIN and elevating LGR5 levels, resulting in the induction of stem cell plasticity in CRC. Our cell line data being consistent with the tumor data is a positive sign that this analysis of the 18-gene and 13-gene scores might be a useful prognostic tool for clinicians down the line. Looking forward, MEKi and SRCi treatments on xenographs with various CMS classes and different 18-gene and 13-gene scores would reveal if our findings are consistent in living organisms. The most definitive finding, though, is a positive 13-gene score with a positive 18-gene score that enhanced apoptotic sensitivity to the combination of Trametinib and Dasatinib, and this drug combination may be effective in the treatment for CRC. Importantly, the biomarkers presented here can predict the subpopulation of tumors that are sensitive to MEKi combined with SRCi. Figure 11 illustrates how the 13-gene and 18-gene scores can be used as a predictive model. The in silico and cell line experiments clearly support our hypothesis that signature scores can select tumors sensitive to MEK and SRC inhibitor combined therapy. This hypothesis, as indicated in Figure 11, awaits confirmational in vivo experimentation.

## 5. Conclusions

MEK inhibitors are a new class of drugs designed to treat RAS mutant CRC. Unfortunately, the effectiveness of these drugs has been limited due to de novo resistance or to the development of adaptive resistance. We identified SRC as a potential Achilles heel in treating MEKi-resistant CRC. We postulated that two gene expression signature scores representing MEKi sensitivity and MEKi adaptive resistance, the latter correlating strongly with SRC pathway activation, might be useful in predicting which CRC subpopulations might respond to the combination of a MEKi and an SRCi. We demonstrated that these signature scores might serve as useful biomarkers to predict in vivo drug sensitivity. Moreover, we determined the same using in silico approaches with both hundreds of cell lines and human tumors. These results lay the groundwork to develop clinically useful biomarker scores to select patients who may be sensitive to this new drug combination in future clinical trials. Moreover, this drug combination might be most effective in the difficult to treat CMS4 class of CRC enriched in stem cells.

## Figures and Tables

**Figure 1 cancers-14-01451-f001:**
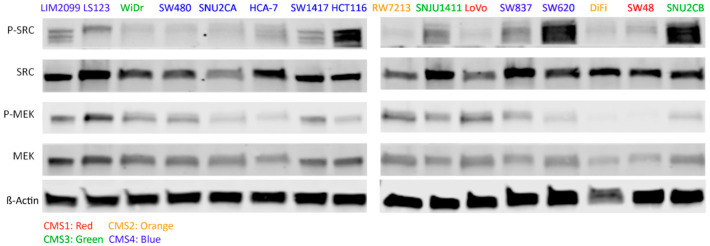
Western Blot Analysis for all 16 cell lines comparing the levels of activated Phospho-SRC Y416 and Phospho-MEK1/2 S217/221. CMS4 (top row of cells plus SW837) do not appear to have significantly higher levels of SRC activation compared to cells of other CMS class. Additionally, most of the activated MEK seems to have similar variation between cell lines regardless of CMS class. All analysis was confirmed by the normalized band density readings featured in the histograms of Appendix A (*n* = 3).

**Figure 2 cancers-14-01451-f002:**
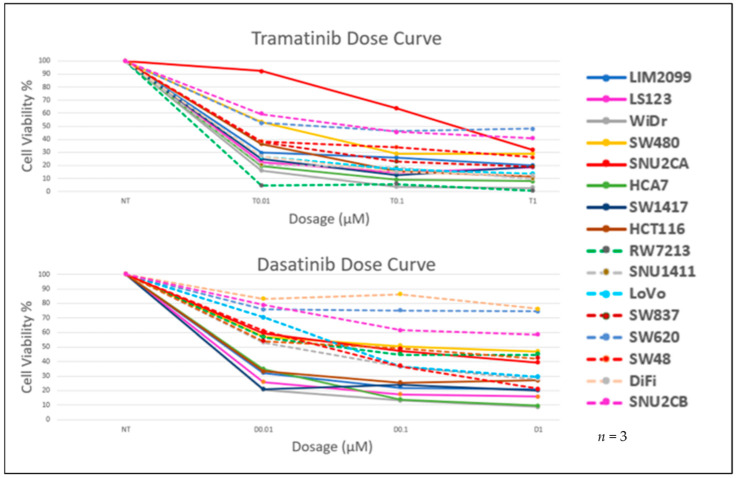
CellTiter-Glo analysis to compare cell dose response curve for Trametinib and Dasatinib for all 16 CRC cell lines used in this study. The assays were performed 96 h post drug treatment with drug concentrations ranging from 0.01 µM–1 µM. Response to both drugs increases as dosing increases. For most of the cell lines, response plateaus at 0.1 µM concentration (showing less than 50% viability), though there are few exceptions (*n* = 3).

**Figure 3 cancers-14-01451-f003:**
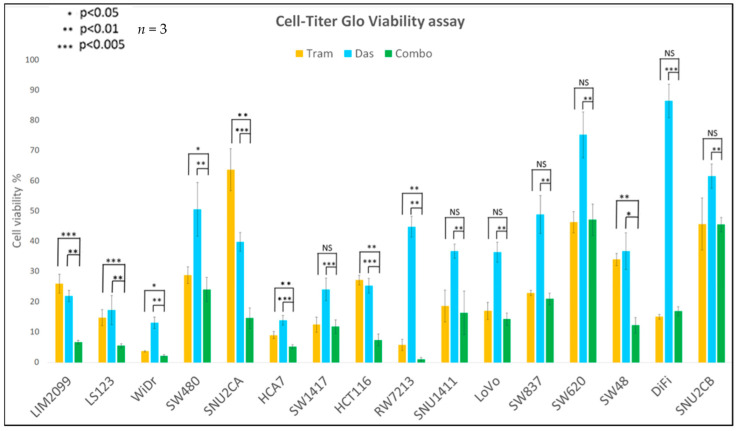
CellTiter-Glo analysis for cell viability percentage of all cell lines, used in this study, treated with Trametinib alone, Dasatinib alone and combination (all 0.1 µM for 96 h). Student’s *t*-test (paired one-tailed) demonstrates a significant decrease in cell viability with use of the drug combination in comparison to using the drugs alone for the following cell lines: LIM2099, LS123, WiDr, SW480, SNU2CA, HCT116 and HCA-7. All other cell lines show nonsignificant decreases with the use of the combination compared to single use Trametinib (*n* = 3).

**Figure 4 cancers-14-01451-f004:**
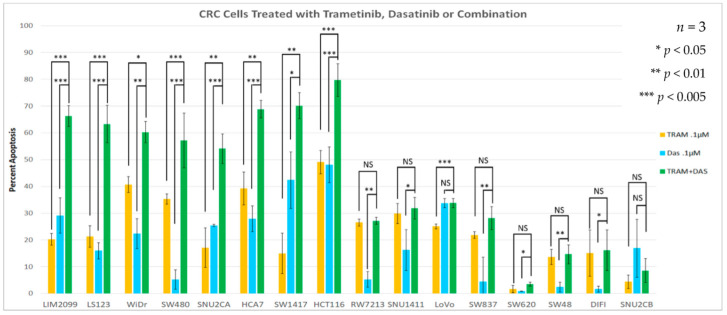
Annexin V analysis for apoptosis of cell lines treated with Trametinib, Dasatinib, and combination (all 0.1 µM for 96 h). Student’s *t*-test demonstrates a significant increase in (singular drug) apoptosis with use of the drug combination for cell lines LIM2099, LS123, WiDr, SNU2CA, SW480, HCA-7, SW1417, and HCT116. All other cell lines prove to have nonsignificant increases with use of the combination (*n* = 3).

**Figure 5 cancers-14-01451-f005:**
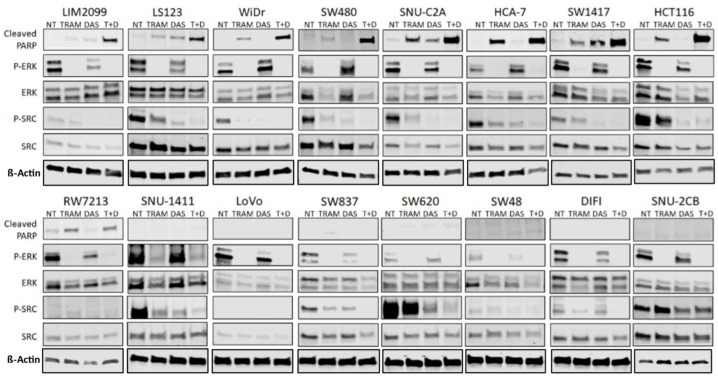
Western Blot analysis of cell lines treated with Trametinib, Dasatinib or combination. Cleaved PARP, Phospho-SRC (Tyr416), SRC, Phospho-ERK1/2 (Thr202/Tyr204), and ERK were probed for. This analysis shows increased cleaved PARP in cells shown to be sensitive with the combination treatment in Annexin V analysis (fig4). Increases in cleaved PARP is shown in all the sensitive cell lines. Inhibitory effects of the drugs are also confirmed with decrease of Phospho-ERK and Phospho-SRC levels in treated cell lines. However, notably some resistant cell lines did not show significant decreases in P-SRC when Dasatinib was used. All findings were confirmed via band density readings featured in the Appendix A histograms.

**Figure 6 cancers-14-01451-f006:**
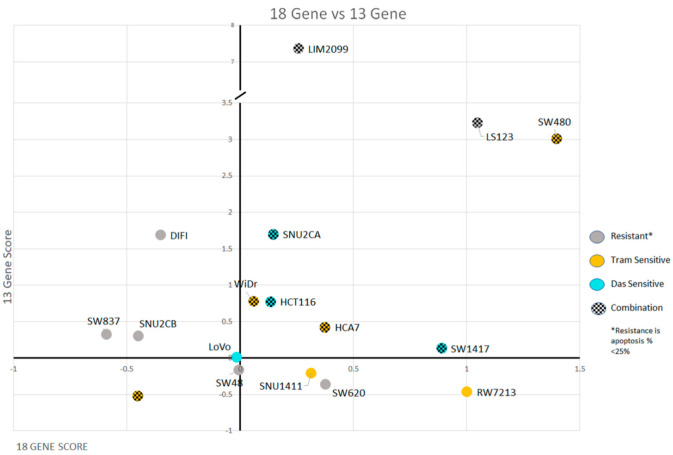
The 18-gene RAS activation score and the 13-gene MEKi Resistance score for the 16 CRC cell lines used in this study were plotted with color indicators for their drug sensitivity. Each cell line is represented as a singular dot that is colored, indicating its apoptotic sensitivity to either Trametinib, Dasatinib, or the combination of both. Cells with the checkerboard pattern are cells with significantly enhanced apoptosis when the combination is used. Notably, 7 out of 9 of the CMS4 cell lines have increased cell death when the combination is used. These cells all have positive 18-gene and 13-gene scores (located in the righthand upper quadrant).

**Figure 7 cancers-14-01451-f007:**
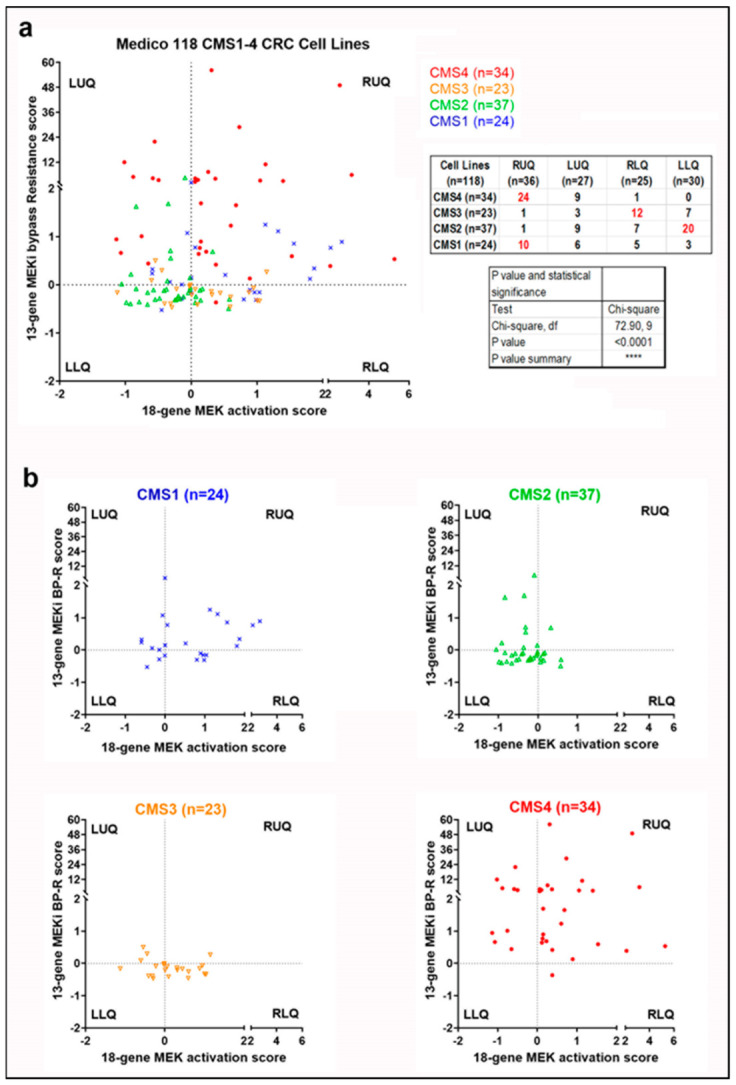
The microarray data from the Medico CRC cell line database were used to calculate and plot the 18-gene and 13-gene signature scores for 118 CMS1-4 CRC cell lines (**a**). CMS status is indicated by color. Here median scores of 18 gene and 13 gene signatures are set as zero. A Chi-square test with a value of 72.9 (*p* < 0.0001) for these cell lines shows a distinctive trend for CMS4 cell lines to have higher 18-gene scores and higher 13-gene scores, which fall in the right upper quadrant (RUQ). Distribution of each CMS class is shown based on what quadrant they fell into (**b**). Nearly all CMS4 cell lines have higher 13-gene scores (locating within primarily RUQ and secondary LUQ). To a lesser degree, the CMS1 cell lines also tend to have higher 13-gene scores. By contrast, the CMS2 and CMS3 cell lines tend to have lower 13-gene scores falling into the lower two quadrants (LLQ and RLQ).

**Figure 8 cancers-14-01451-f008:**
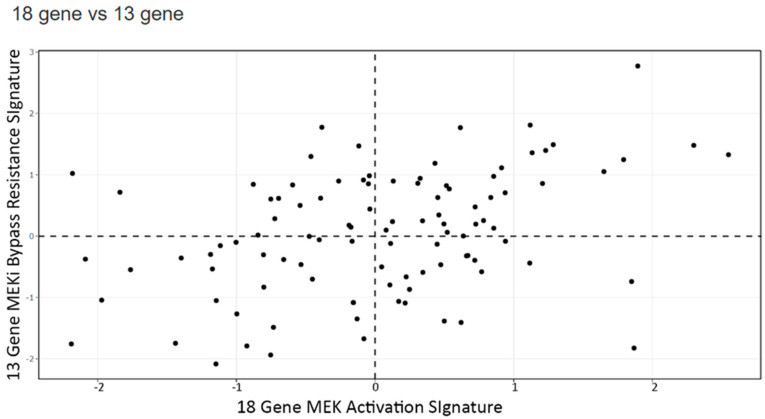
A scatter plot for the 101 human CRC tumors of published RNAseq data-set was used to score 101 human CRC tumors for their 18-gene and 13-gene scores.

**Figure 9 cancers-14-01451-f009:**
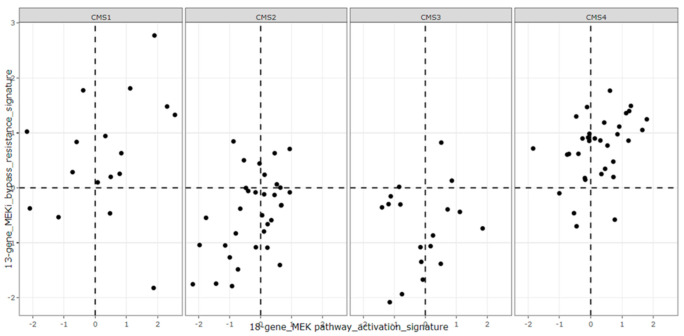
The 101 human CRC tumors from Figure 6 were broken down by CMS category and plotted using their 18-gene and 13-gene scores. CMS4 tumors show the highest number of tumors with positive 18-gene and 13-gene scores with CMS2 and CMS3 showing distinctively fewer tumors with high 13-gene scores.

**Figure 10 cancers-14-01451-f010:**
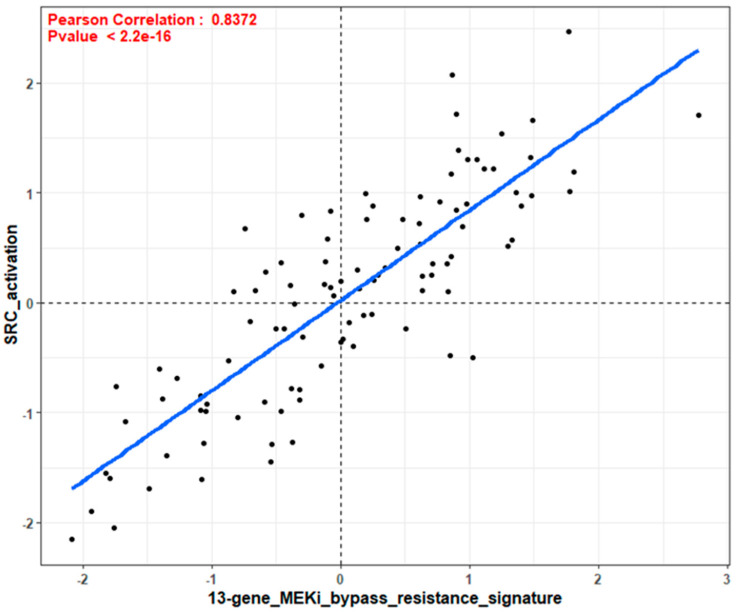
Using the RNAseq Vasaikar et al. database, 101 human tumors were plotted for the SRC activation score and 13-gene score. Plotting these two scores against one another shows a strong correlation with a Pearson Correlation value of 0.8372 and a *p* value of <2.2 × 10^−16^. This correlation suggests that high 13-gene is indicative of high Src activation.

**Figure 11 cancers-14-01451-f011:**
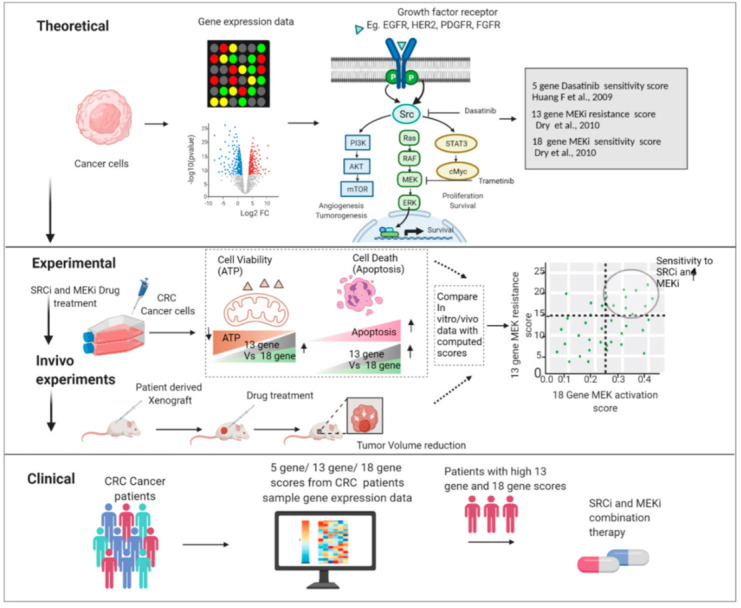
Early phase translational strategies to develop selective effective clinical treatment for CRC. Theoretical setting: publicly available cancer gene expression datasets were used for gene signature scores specifically generated to determine the activity of the RAS/MEK pathway and the resistance to MEK inhibition. Experimental validation: the theoretically calculated gene signature scores specifically for colon cancer were evaluated in colorectal cancer cell lines from each CMS group to analyze Src inhibitor and Mek inhibitor on cell viability and cell death. Proposed Clinical: In vivo trials of our theoretical and experimental findings can lead to patient selection with 13-gene MEK resistance score and 18-gene RAS pathway score for clinical treatment with the combination of SRCi and MEKi.

**Table 1 cancers-14-01451-t001:** Molecular characteristics and signature scores of 16 CRC cell lines used.

Cell Lines	Cancer Characteristics			Scores		
	MSI/MSS Status	Mutation Gene	CMS Class	18 Gene	13-Gene	SRC Activation
LIM2099	MSS	KRAS	CMS4	0.26	7.36	6.78
LS123	MSS	KRAS	CMS4	1.05	3.22	1.77
SW480	MSS	KRAS	CMS4	1.40	3.01	1.08
SW1417	MSS	BRAF	CMS4	0.89	0.13	1.22
SNU2CA	MSI	KRAS	CMS4	0.15	1.70	0.51
HCT116	MSI	KRAS	CMS4	0.14	0.77	−0.17
WiDr	MSS	BRAF	CMS3	0.06	0.78	1.10
HCA7	MSI	WT	CMS4	0.38	0.42	0.20
SW620	MSS	KRAS	CMS4	0.38	−0.36	−0.18
SNU2CB	MSI	KRAS	CMS3	−0.45	0.30	−0.42
SNU1411	MSS	KRAS	CMS3	0.32	−0.21	1.45
DIFI	MSS	WT	CMS2	−0.35	1.69	1.14
LoVo	MSI	KRAS	CMS1	−0.01	0.01	0.07
SW48	MSI	WT	CMS1	0.00	−0.17	−0.45
SW837	MSS	KRAS	CMS4	−0.59	0.32	−0.17
RW7213	MSS	KRAS	CMS2	1.00	−0.47	−0.41

## Data Availability

Any datasets used in this paper have been cited and are publicly accessible.

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
