# Peer review of "Ras Pathway Activation and MEKi Resistance Scores Predict the Efficiency of MEKi and SRCi Combination to Induce Apoptosis in Colorectal Cancer"

_cancers, 2022, doi:10.3390/cancers14061451_

Round 1
Reviewer 1 Report
Dear authors,
Thanks for laying out a well written manuscript reflecting hard and persistent work.
Below are some points to be addressed
Line 165: Was determined
Please check article for similar editing
Figure 1:
It is a bit hard to follow up and accept the conclusion that "CMS4 (top row of cells plus SW837) do not appear to have significantly higher levels of SRC activation compared to cells of other CMS class"
To validate above statement please quantify the bands using a suitable software. Then please plot the outcome with bar graphs and suitable scales to be able to visually compare the expression level of p-SRC. Kind reminder to normalize P-SRC over total SRC and not actin as it is crucial to include the variation of the total SRC protein expression in the analysis of any conclusion or statement
No indication of sample size (n). Please indicate your sample size
Figure 2:
Please indicate sample size (n)
Figure 3:
Line 246 The statement (All other cell lines show nonsignificant decreases with the use of the combination) is misleading as the combo is significantly lower than Das for most cell lines
Please indicate sample size for all experiments
Thanks.
Author Response
The manuscript has been edited again for grammatical errors.
All bands in figure 1 have been quantified and normalized as suggested. P-values for the cell lines have also been presented. Histograms were made for the westerns in Figures 1 and 5 and have been added to the supplemental information (Figure S1 and S3).
Sample size n=3 has been added to the figures and legends for all experimental work as it was all done in triplicate at the very least.
Figure 3 has been clarified to state “All other cell lines show nonsignificant decreases with the use of the combination compared to single use trametinib.” The purpose of this experiment ultimately was to see if the addition of Dasatinib to cells treated with Trametinib would create enhanced efficacy to the baseline Trametinib activity. As it appears 8 cell lines show more cell growth when Dasatinib is used than when Trametinib is used. So, the comparison likely should just be between Trametinib and the combination, but we needed to confirm and present that Dasatinib was not the dominating factor in preventing cell growth. Except for SNU-2CA, all Dasatinib treatments across the cell lines proved to have greater than or equal to the cell growth of Trametinib treatments. Furthermore, with certain cells we see that the Dasatinib has an additive effect to Trametinib treatments in preventing cell growth and pushing apoptosis.
I hope that these changes address the issues brought forward with the manuscript. Addressing these points has strengthened the paper overall. Thank you for your time and consideration.

Reviewer 2 Report
In this manuscript the Authors hypothesized that the MEK inhibitors resistance in colorectal carcinoma (CRC) may be due to SRC activation and that inhibition of SRC may suppress MEK inhibition resistance mechanisms. They evaluated targeting two pathways simultaneously to produce an effective treatment by overcoming resistance, combining Trametinib with Dasatinib and provides enhanced cell death in cell lines system. Moreover, they found that the sensitive cells lines to the dual treatment were predominantly CMS4 class of CRC enriched in stem cells. An 18 gene RAS pathway signature score and 13 gene MEKi resistance signature score have be showed as biomarkers for define CRC cells sensitivity to combination of Trametinib with Dasatinib, both in in vitro and in silico experiments.
Comments:
- All Figures in the text which include western blot analysis have to comprise a histogram of the results and a p-values have to be indicated.
- The Authors should better explain how they obtained the 18 gene RAS pathway score and the 13 gene MEKi resistance score in in vitro and in silico experiments.
Author Response
All western blots have added in histograms with p-value scores. For Figure 1 p values between LIM2099 and all subsequent cell lines were performed as doing all permutations of every cell line would be difficult to present visually.
These histograms were added into supplementary data as not to hurt the existing image quality. Figure 5 already has a lot of information in it and adding in 3 very complex histograms would detract from the presentation.
Additionally, ß-Actin bands were added to Figure 5 due to Cleaved PARP being normalized with ß-Actin.
As far as how the 18 gene RAS activation score and 13 gene MEKi Resistance score were created, these scores come from separate papers both of which are cited in the text (Dry and Broecker). This explanation is presented in the methods section (2.5). Another paragraph with a bit more detail was added to this section of the methods to explain how these scores were used in our analysis specifically. How they determined the scores is a bit complicated. However, these scores have been cited by many times and used in multiple publications to varying degrees.
I hope that these revisions were able to address the points you put forward. Thank you for your time and consideration in reviewing this manuscript.

Round 2
Reviewer 2 Report
The authors replied comprehensively to all comments